# A Displacement Controlled Fatigue Test Method for Additively Manufactured Materials

**Mohammad Masud Parvez** [1,*] **, Yitao Chen** [1] **, Sreekar Karnati** [1] **, Connor Coward** [1] **, Joseph W. Newkirk** [2] **and Frank Liou** [1]

[1]  Department of Mechanical and Aerospace Engineering, Missouri University of Science and Technology, Rolla, MO 65401, USA

[2]  Material Science and Engineering, Missouri University of Science and Technology, Rolla, MO 65401, USA

*  Correspondence: mphf2@umsystem.edu; Tel.: +1-573-202-1506

**Abstract:** A novel adaptive displacement-controlled test setup was developed for fatigue testing on mini specimens. In property characterization of additive manufacturing materials, mini specimens are preferred due to the specimen preparation, and manufacturing cost but mini specimens demonstrate higher fatigue strength than standard specimens due to the lower probability of material defects resulting in fatigue. In this study, a dual gauge section Krouse type mini specimen was designed to conduct fatigue tests on additively manufactured materials. The large surface area of the specimen with a constant stress distribution and increased control volume as the gauge section may capture all different types of surface and microstructural defects of the material. A fully reversed bending $(R = -1)$ fatigue test was performed on simply supported specimens. In the displacement-controlled mechanism, the variation in the control signal during the test due to the stiffness variation of the specimen provides a unique insight into identifying the nucleation and propagation phase. The fatigue performance of the wrought 304 and additively manufactured 304L stainless steel was compared applying a control signal monitoring (CSM) method. The test results and analyses validate the design of the specimen and the effective implementation of the test bench in fatigue testing of additively manufactured materials.

**Keywords:** adaptive control; fatigue testing; simply supported bending; mini specimen; additive manufacturing; 304L stainless steel

## 1. Introduction

Fatigue is a progressive and permanent structural change due to fluctuating stresses or strains subjected to a material. 50% to 90% of mechanical failures of structures are due to fatigue [1,2]. Fatigue test is indispensable in the characterization of materials but the test is both time-consuming and very expensive [3,4]. In this research, a unique test setup was designed and developed to reduce the test cost using mini specimen. The measured strength of a material subjected to monotonic or cyclic loading depends inversely on the specimen size. The impact of the size effect on mechanical properties depends on the type and local feature of the material structure i.e., grain size, microcracks, inclusions, discontinuities, dislocations, and other defects [5–7]. Extended studies were carried out to investigate the effect of specimen size and loading condition on fatigue behavior of metallic materials [8–15]. Statistically, large specimens contain more extreme defects. The presence of larger defects leads to crack growth and failure at lower stress levels. Sun [16] proposed a probabilistic method to correlate the effects of specimen geometry and loading condition on the fatigue strength based on the Weibull distribution. Tomaszewski [4] performed comparative tests on mini specimens and normative specimens, and verified the monofractal approach based on Basquin's equation along

with the Weibull weakest link model. There are some other statistical methods proposed to evaluate the size effect on the fatigue test [17–21]. All of these approaches epitomize that standard specimens demonstrate lower fatigue strength than mini specimens due to the higher probability of larger material defects. Additively manufactured materials have a higher probability of defects compared to wrought materials. In this paper, the implementation of a dual gauge section Krouse type mini specimen increases the surface area to capture all different types of surface and microstructural defects since most of the fatigue failures are initiated at the surface or subsurface due to the presence of defects.

Geometrically, the size effect is related to the nonlinear distribution of the stress [22–24]. The stress gradient occurring under bending and shear stress has a higher influence on the size effect for a bending type test compared to axially loaded cyclic test but the axial fatigue test on mini specimens suffers buckling. In this study, the transverse bending test with a constant stress distribution within the gauge section in a specimen eliminates the stress gradient effect.

There are several techniques already developed to monitor the crack nucleation and propagation during the fatigue test. These techniques include the acoustic emission diagnostic method [25–27], electrical resistance change method [28,29], meandering winding magnetometer (MWM)-array eddy current sensing [30], and thermographic method [31]. All of these techniques require an additional sensor with intensive signal processing. In the current work, we introduce a simple but effective control signal monitoring (CSM) method to identify the nucleation and propagation phase. In a displacement-controlled mechanism, the control signal decreases with the decrease in the structural stiffness of the specimen. The change in the control signal provides insight in estimating the nucleation and propagation phase. In this study, the fatigue test was conducted on wrought 304 and additively manufactured 304L stainless steel specimens. The CSM method was applied to identify the nucleation and propagation phase. The test results were compared to validate the design of the specimen and the test setup performance in high cycle fatigue testing.

## 2. Methodology

In this study, a fully reversed bending (R = −1) fatigue test was performed on simply supported specimens. A simply supported testing methodology has several advantages over a fully clamped type of loading mechanism. The maximum deflection in a simply supported and a fully clamped beam with a concentrated load *F* at the center are given by Equations (1) and (2) respectively [32,33],

$$\delta_{max} = \frac{Fl^3}{48EI} \tag{1}$$

$$\delta_{max} = \frac{Fl^3}{192EI} \tag{2}$$

where *F*, $\delta_{max}$, *l*, *E*, and *I* are the applied force, maximum deflection, length, modulus of elasticity, and moment of inertia of the beam respectively. For a given load, the displacement is four times higher in a simply supported bending than in a fully clamped bending. During the fatigue test, investigators attempt to actuate the specimen at its natural frequency to achieve maximum displacement. However, the dynamics of the actuator coupled with the specimen limit the operation. Therefore, as an alternate, we adopted a simply supported bending mechanism as the testing methodology.

## 3. Specimen Design, Analysis and Preparation

### 3.1. Design of the Specimen

A dual gauge section Krouse type mini specimen was designed for simply supported loading. The specimen is a modified form of the ASTM (American Society for Testing and Materials) International standard B593-96(2014)e1, definition E206, and practice E468 [34]. Some authors already reported on the modification and implementation of the specimen in miniature form [35–38]. Since the specimens are miniature size, Haydirah [39] performed an error analysis based on the effect of

specimen's dimension. Figure 1 shows the dimension of our specimen. The effective length between both clamping ends is 25.4 mm. Each gauge is 4.34 mm long. The total gauge covers 34.17% of the total effective length of the specimen. The dual gauge increases the overall surface area. The failure is expected to be within the gauges. Another reason for choosing the dual gauge is to maintain symmetry. In a single cantilever beam, the actuator follows a curved path during excitation. To keep the path of the actuator one dimensional, and to distribute the load symmetrically along with the specimen, the dual gauge concept is opted.

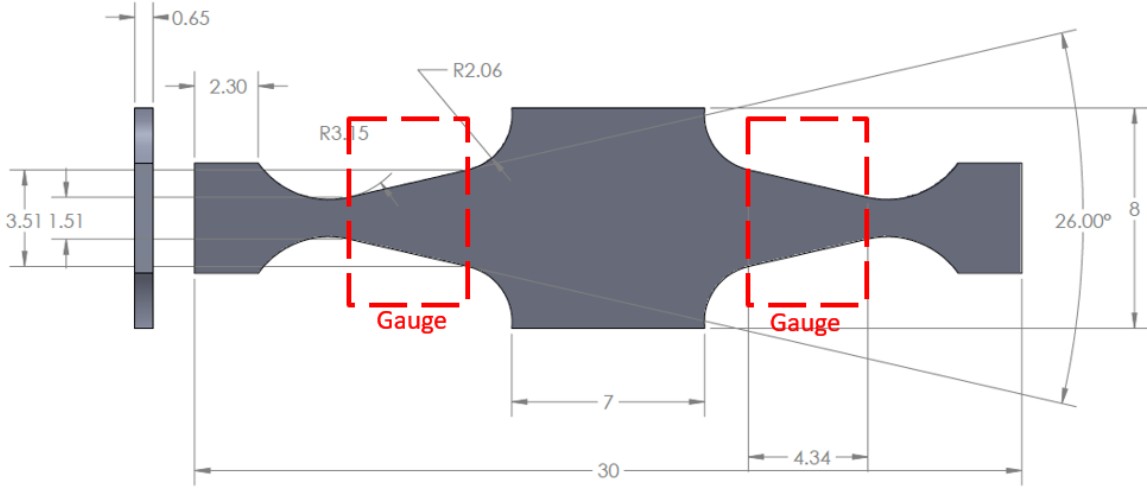

**Figure 1.** Drawing of the dual gauge section Krouse type mini specimen, all units are in mm.

### 3.2. Stress Calculation

Previous studies showed that simple beam equation is applicable to calculate the stress in miniature wedge shaped specimen [34–39]. The stress in a simply supported bending beam with a point load at the center can be expressed as [40],

$$\sigma = \frac{M(x)}{I(x)} \frac{h}{2} \tag{3}$$

where, $\sigma$, $M(x)$, $I(x)$, and $h$ are the stress, moment, second moment of inertia, and the thickness of the specimen, respectively. For a simply supported beam, $M(x) = \frac{Fx}{2}$, and $I(x) = \frac{b(x)h^3}{12}$ where, $F$, and $b$ are the point load, and the width of the specimen, respectively. For a Krouse type specimen $b(x) = 2kx$ where, $k$ is the slope of the specimen. Inserting $M(x)$, and $I(x)$ in Equation (3), we get,

$$\sigma = \frac{3F}{2kh^2} = j(F, h) \tag{4}$$

where, $j$ is the stress function. The nominal stress $\sigma$ within the gauge in Equation (4) depends on the force applied and the thickness of the specimen, not on the distance $x$. Ideally, a constant stress distribution is expected but in reality at the defect zone or at the lower strength site, the actual local stress will be higher than the nominal stress.

### 3.3. Sensitivity and Uncertainty Analysis

The specimen is a miniature size compared to the standard one. The necessity of sensitivity and uncertainty analysis is inevitable to determine the optimal thickness of the specimen. The stress calculation is sensitive to the force and thickness of the specimen according to Equation (4). Uncertainty in force measurement depends on the sensor's accuracy, calibration, and set up. The thickness is sensitive to the machining and polishing process. For a higher thickness, a higher force is required to

attain particular stress. This leads to the necessity of a high power system and actuator. An optimal thickness was determined to eliminate the necessity of high power fatigue machine and external cooling. Partially differentiating Equation (4) we get,

$$\nabla \bar{j} = \begin{bmatrix} \frac{F}{\sigma} \times \frac{\partial \sigma}{\partial F} \\ \frac{h}{\sigma} \times \frac{\partial \sigma}{\partial h} \end{bmatrix} = \begin{bmatrix} 1 \\ -2 \end{bmatrix} \tag{5}$$

From Equation (5), we can see 1% variation in specimen thickness produces 2% change in stress value. To estimate the thickness uncertainties, 10 specimens were prepared. The thickness was measured using a high precision laser displacement sensor. The uncertainty was calculated obtaining overall standard deviation (std) using Equation (6).

$$std = \frac{1}{n} \sum_{j=1}^{n} (x_j - \bar{x})^2 = \frac{1}{n} \left[ \sum_{i=1}^{g} n_i S_i^2 + \sum_{i=1}^{g} (\bar{x}_i - x)^2 \right] \; textrmwhere, \bar{x} = \frac{\sum_{i=1}^{g} n_i x_i}{n} \tag{6}$$

where, $\bar{x}_i$, $S_i$, and $n_i$ are the mean, standard deviation, and the number of scanned data points of $i$ th specimen respectively. $\bar{x}$ is the overall mean, and $n$ is the total number of data points. For $\pm 5\%$ stress variation, the calculated optimal thickness of the specimen was 0.509 mm with three sigma quality level. Including a factor of safety, the specimen thickness used in this study is 0.65 mm.

### 3.4. Finite Element Analysis

Finite element analysis was performed using ABAQUS 2018 software (Dassault Systèmes Simulia Corp; Providence, RI, USA) to demonstrate the constant stress distribution within gauge sections. According to the specimen design, as shown in Figure 2, the 3D prototype of the specimen was simply supported at both sides which are marked by red lines ($U_z = 0$). To ensure a symmetric deformation, the displacement on center-lines along the $x$ -axis (green line) and $y$-axis (blue line) are restricted in y direction ($U_y = 0$) and x direction ($U_x = 0$), respectively. A constant displacement $U_z = 0.150$ mm was applied on the 3 mm $\times$ 7 mm dark grey rectangular area at the center of the specimen, which indicates the rectangular plate washer in the machine setup. Boundary conditions are listed in the box under the 3D prototype of the specimen. The Young's modulus and Poisson's ratio set for the wrought 304 stainless steel were 200 GPa and 0.3 respectively. A linear elastic model was applied to observe the mechanical response under this static condition, as the deformation is within the elastic regime. The distribution of the nominal stress S11 on the whole specimen is then obtained by the simulation, as shown in Figure 3. A constant nominal stress within triangular gauge sections can be observed, and it reaches the maximum value at the surface. Convergence study was also performed by selecting 6 different mesh sizes which result in the number of elements ranging from 1263 to 166,506. The data points in Figure 4 shows that the nominal stress converges to approximately 177.7 MPa as the number of elements increases to 166,506, since when the number of mesh elements increases from 76,698 to 166,506, the change in nominal stress value is less than 0.2%. Figure 3 exhibits the nominal stress distribution with the number of elements of 166,506. The sole purpose of using FEA analysis is to demonstrate the stress distribution within the gauge.

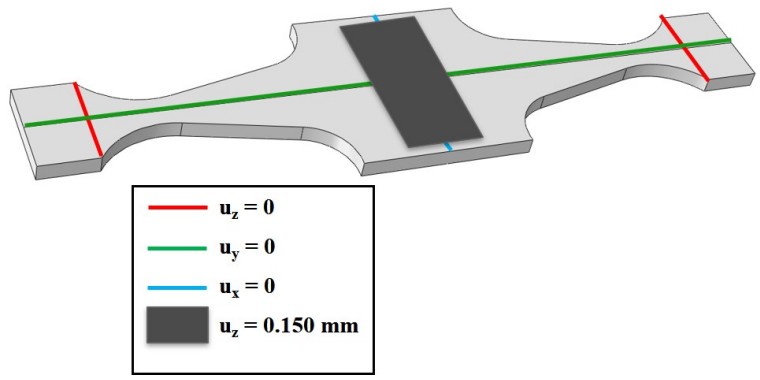

**Figure 2.** FEA simulation setup for wrought 304 stainless steel specimen.

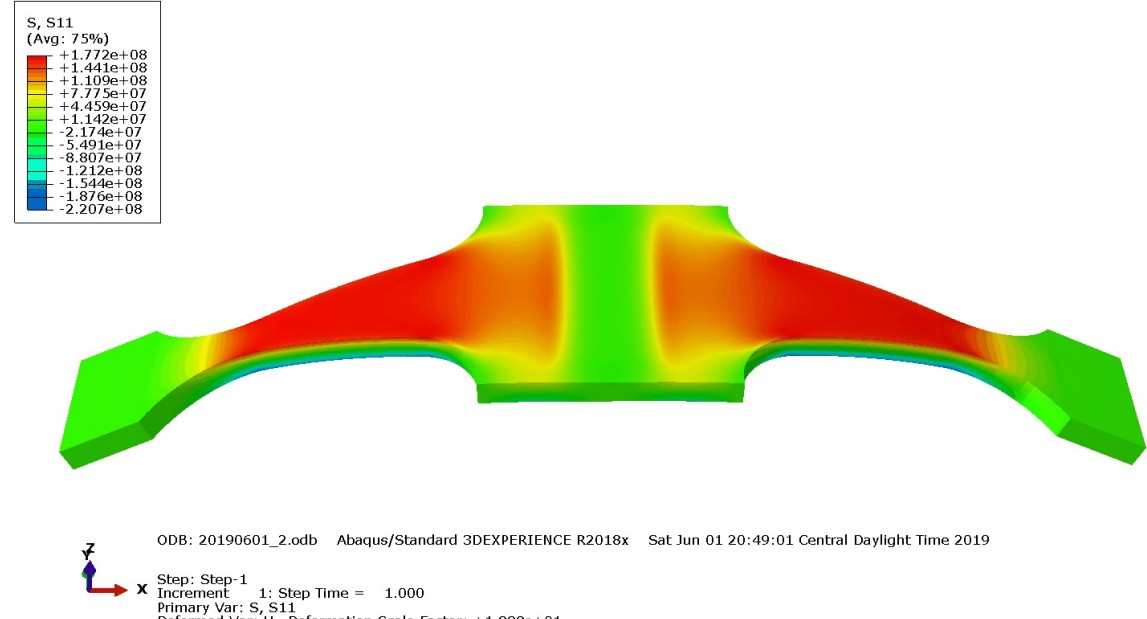

**Figure 3.** FEA simulation result of the specimen. The red zone on the top surface shows the stress distribution is constant within the gauges.

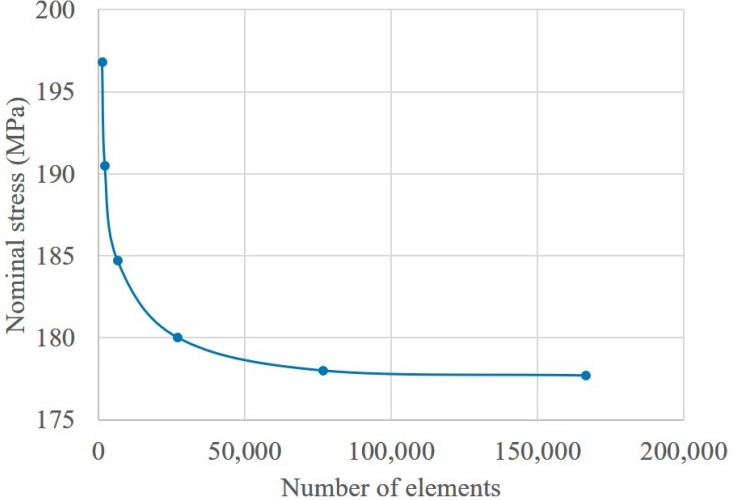

**Figure 4.** Convergence analysis of the FEA simulation results.

### 3.5. Materials and Specimen Preparation

The materials tested in this study are hot rolled and annealed 304 stainless steel bulk material and additive manufacturing (AM) fabricated 304L SS bar. These materials were chosen because they are economical and widely used due to their strength and high resistance to corrosion. The chemistry of both the wrought material and powder used as the feedstock for AM is listed in Table 1. The relatively close chemistry of both materials except Ni which is 2% higher but not expected to make a significant difference in the test results may aid a better understanding of the comparative study. Rough finish (average $R_a$ = 3.82 μm) and fine finish (average $R_a$ = 0.482 μm, average $R_z$ = 4.242 μm) wrought specimens were machined using W-EDM, while additively manufactured fine finish specimens were cut along $Z$ axis from a bar fabricated using the selective laser melting (SLM) process. A Renishaw AM250 machine was used to build the part. An optimal process parameter listed in Table 2 was applied to yield maximum part density. A total of 10 specimens for each type were manufactured with no additional surface preparation.

**Table 1.** Chemical properties (wt%) of 304L stainless steel powder and bulk 304 stainless steel.

| Material | C | Mn | Si | S | P | Cr | Ni | Cu | Mo | Co | N | O |
|---|---|---|---|---|---|---|---|---|---|---|---|---|
| Wrought | 0.023 | 1.69 | 0.43 | 0.020 | 0.034 | 18.10 | 8.02 | 0.63 | 0.24 | 0.15 | 0.084 | - |
| Powder | 0.015 | 1.40 | 0.63 | 0.004 | 0.012 | 18.50 | 9.90 | <0.1 | - | - | 0.090 | 0.02 |

**Table 2.** Parameters used to build additively manufactured part using selective laser melting (SLM) process.

| Parameter Set | Power (watt) | Hatch Space (μm) | Point Distance (μm) | Exposure Time (millisecond) | Energy Density (MJ/m$^3$) | Raster Rotation (degree) |
|---|---|---|---|---|---|---|
| Nominal | 200 | 85 | 60 | 75 | 58.8 | 67 |

## 4. Experimental Setup

### 4.1. Mini Fatigue Testing Machine

The mini fatigue testing machine consists of six major parts: (i) an electromagnetic actuator, (ii) a non-contact displacement sensor, (iii) a load cell, (iv) a controller, (v) a power amplifier or driver, and (vi) a test bench. The voice coil of a subwoofer was used as the actuator. The sub-woofer behaves as a low audio frequency shaker. The mathematical model of an electrodynamic shaker and a sub-woofer is relatively similar though the moving elements of a shaker are more rigid than a subwoofer. Higher rigidity multiplies the power requirements. To design a low power system, a soft mechanical suspension of the sub-woofer was implemented to transfer maximum energy to the specimen. The actuator is made of a high Curie temperature ferrite magnet with a cast aluminum frame of 10 inches diameter. The larger diameter of the voice coil (3 inches) than the length (1 inch) of the specimen supports the one-dimensional movement. The dust cap of the voice coil was replaced with a plastic flange. A load cell was mounted in-line between the central clamp and the flange to measure the tensile and compressive force. To measure the displacement of the specimen, a high-speed non-contact laser displacement sensor was fixed with a guide rail. Figure 5 illustrates the test bench setup. First, the specimen was clamped at the center. The specimen sits on the bearings at both ends as shown in Figure 5. Spacers were used at both ends to ensure no preload on the specimen. Then, the other bearing holders were placed and clamped using heavy load toggle clamps on top. To ensure line contact at both ends, bearings were used. Bearings also minimize the friction during the simply supported vibration test. By sliding the displacement sensor using the guide rail, the sensor was pointed at the center of the specimen. The displacement measured by the sensor was processed using a microcontroller to determine the amplitude and mean of the displacement. The data was sent to a computer from the microcontroller using a serial port. An adaptive controller was implemented in

the Python development environment to estimate the required amplitude of the sinusoidal control signal. The signal from the computer was sent to a waveform generator via Ethernet. A linear power amplifier connected with the waveform generator drives the actuator. All process and manipulated variables were stored for further analysis to identify the nucleation and propagation phase.

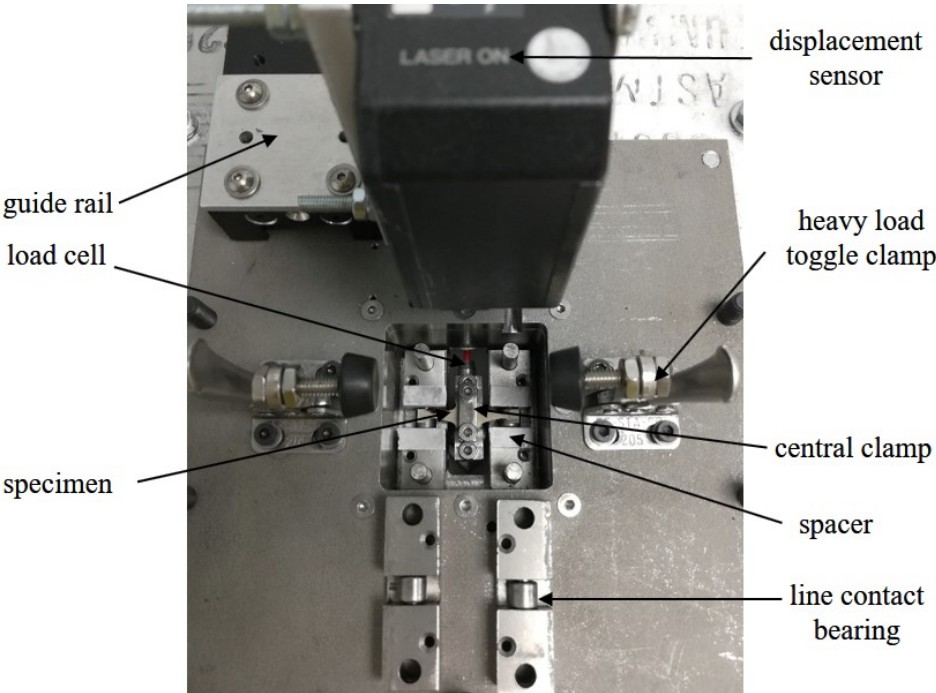

**Figure 5.** Fatigue test bench with a specimen mounted.

### 4.2. Adaptive Controller Design

An adaptive proportional and derivative (PD) controller was designed to control the displacement amplitude. A conventional PID controller was avoided since the system parameters change due to the structural stiffness change of the specimen during the test. Material hardening or softening may occur too. There may also be a possibility of overshoot above the set point during the transient condition. Overshoot may affect the test results. Therefore, an adaptive controller was designed. The design of an adaptive controller follows Equation (7).

$$u(k) = u(k-1) + P * error + D * \frac{error(k) - error(k-1)}{\Delta t} \tag{7}$$

where, $u(k)$, and $u(k-1)$ are the control signal amplitudes at time $k$, and $(k-1)$ respectively. $P$, and $D$ are the proportional and derivative gain respectively, and $\Delta t$ is the time step. The proportional controller offsets the current value linearly with the error, and the derivative controller adds in controlling the actuation based on the rate of the change of error. The error is defined as,

$$error = d_{p-p}^{set} - d_{p-p}^{current} \tag{8}$$

where, $d_{p-p}^{set}$ and $d_{p-p}^{current}$ are the desired and current displacement amplitude respectively. The controller values were chosen by manual tuning with caution that no overshoot occurs above the set point. The controller values need to be varied with the test frequency and test material as well. The $P$ and $D$ controllers were set at 5.0 and 0.5 respectively for 304 materials at 56 Hz test frequency.

## 5. Results and Discussion

The closed-loop displacement-controlled fatigue test was performed on wrought 304 and SLM fabricated 304L SS specimens. The sinusoidal excitation frequency was set at 56 Hz. Figure 6 shows

the displacement amplitude of a specimen actuated at 0.100 volts control signal amplitude for different frequencies. The system response is a window function with 39 Hz cutoff frequency. In terms of system dynamics, the fatigue test is a harmonic forced vibration of two mass-spring systems. One is the actuator, and another is the specimen. At cutoff frequencies, the system behaves like a shock absorber. A detailed explanation can be found in the literature [41]. The frequency response shows that the displacement amplitude is maximum at 56 Hz, 95 Hz, and 134 Hz. The test frequency was chosen 56 Hz since external cooling may be required at higher frequencies. All experiments were conducted for simply supported fully reversed bending test at room temperature. During the test, the temperature of the specimen was monitored using an infrared temperature sensor. The deviation in the temperature remains within ±2 °C at 56 Hz test frequency.

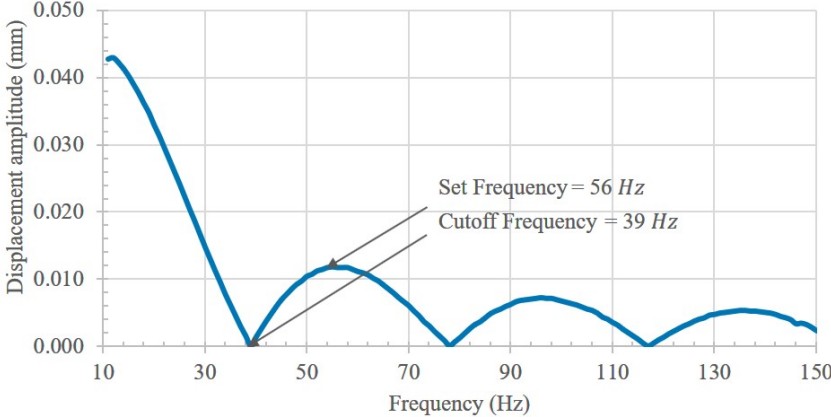

**Figure 6.** The frequency response of a specimen actuated at 0.100 volts control signal amplitude.

In a Krouse type specimen, the fatigue failure can occur at any location within the gauge. All the specimens tested in this study failed within the gauge as expected. The random failure location is due to the defects present randomly within the gauge. The nominal stress distribution is supposed to be constant while the local stress is expected to be high at the defect zone. Figure 7 exhibits the failure location of wrought specimens. Figure 8 illustrates the displacement and control signal amplitude for the wrought specimen actuated at 0.200 mm amplitude which corresponds to 514.26 MPa nominal stress. During the test, the stiffness of the specimen decreases as the crack grows, propagates, and final failure occurs. The control signal amplitude decreases with the reduction of stiffness to maintain the desired set displacement. The displacement amplitude increases suddenly during the final failure. The test was stopped automatically when the amplitude was above a threshold. The test result at different displacements illustrated in Figure 9 validates the effective performance of the adaptive controller.

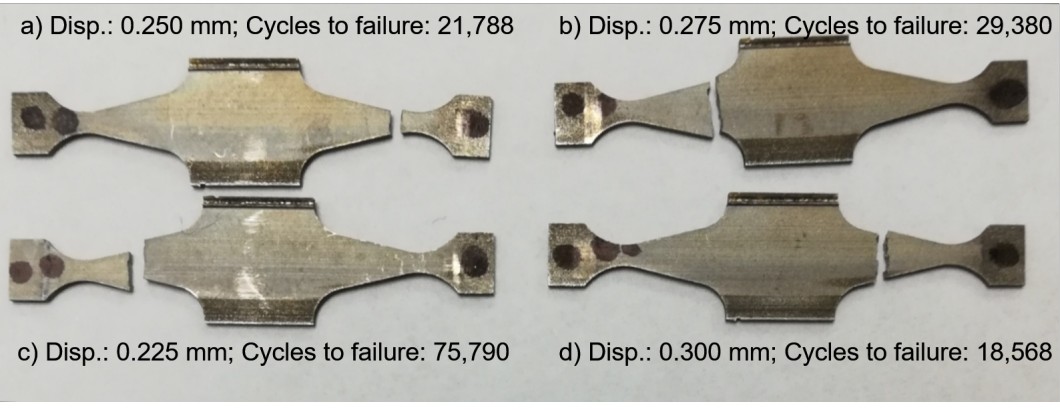

**Figure 7.** Fatigue failure of specimens actuated at different displacement amplitude.

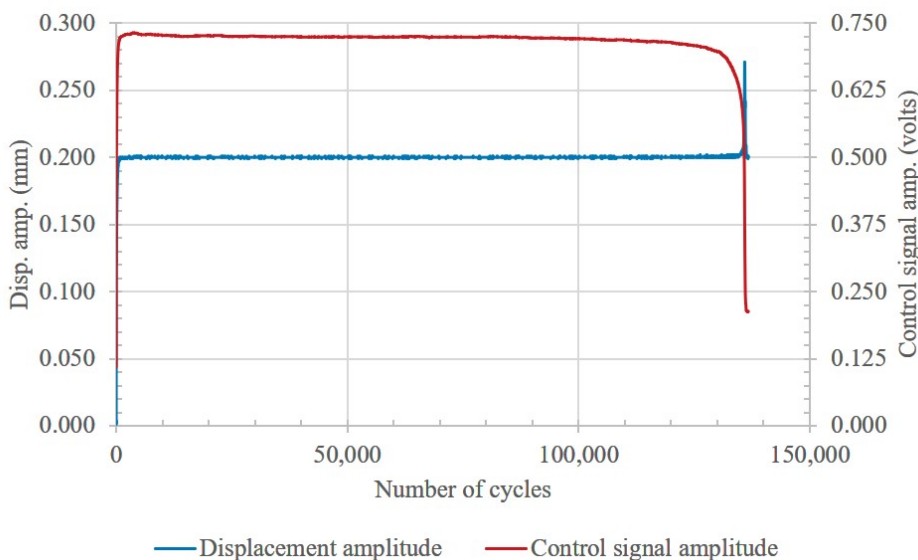

**Figure 8.** Displacement and control signal amplitude up to the entire fatigue life cycle of a fine finished wrought specimen.

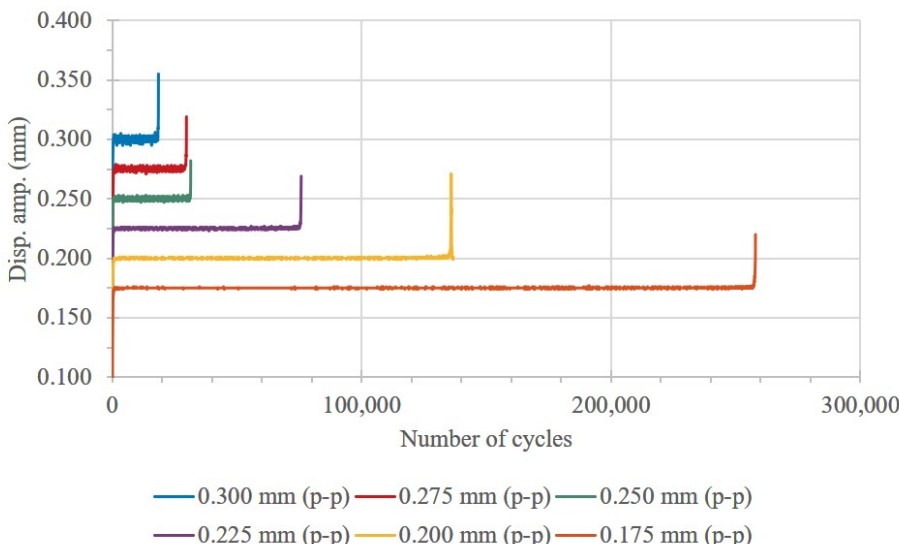

**Figure 9.** Displacement amplitude control of fine finished wrought specimens up to the entire fatigue life cycle.

In this study, we introduce a control signal monitoring (CSM) method to identify the nucleation and propagation stage. During the fatigue test, the crack first grows slowly, which is termed as nucleation. When the nucleation process ends, the crack starts propagating and the final failure occurs. The stiffness of the specimen decreases at all stage, but the rate of the stiffness change is different. Wang [42] reported for carbon fiber polymer-matrix composite that the stiffness of the material has an inverse analogous relation with the change in resistance up to the end of nucleation phase while performing the electrical potential technique on fatigue test to identify the nucleation and propagation phase. Grammatikos [31] implemented the linear regression analysis on the relative potential change as a function of fatigue life fraction to identify different stages. Similarly, this is possible to identify the phases using linear regression analysis on the control signal. First linear regression was applied on the control signal. Then the peak amplitude of the signal near the end of regression line was marked as the end of nucleation phase, as shown in Figure 10. Here the regression helps in choosing the peak of the signal. Current study demonstrates the implementation of CSM method. Future study

includes the sensitivity analysis on the monitoring signal. The implementation of the CSM method on other wrought and SLM specimens is shown in Figure 11. To determine the maximum nominal stress, the average of the peak load was calculated up to the end of the nucleation phase. Inserting the average in Equation (4), the nominal stress was calculated. Figure 12 shows the tensile and compressive load response up to the final failure for the wrought specimen displaced at 0.200 mm amplitude.

The fatigue life of a specimen in terms of the number of cycles is the sum of cycles during nucleation and propagation stages. Materials demonstrate a higher life cycle at low-stress value. Figure 13 shows the end of the nucleation phase cycle and cycles to failure for fine finished wrought and SLM specimens. The trend of nucleation and propagation presented in the literature [43,44] supports the results. Both the wrought and SLM specimens demonstrate an increase in the nucleation and propagation cycle as the stress goes low, but for a particular stress value, both the nucleation and propagation cycles for the SLM material is lower than those for the wrought material. A possible reason is that additively manufactured materials have a higher probability of different types of defects. The fracture surface analysis of wrought and SLM specimens is shown in Figure 14. In the SLM specimen, the crack is initiated at the lack of fusion defect near the top surface while surface defect is the crack initiation source for wrought specimens. We observed similar type of defects to be the crack nucleation site for other SLM specimens. The crack starts growing earlier in SLM materials than in wrought materials. Therefore, the nucleation life cycle is less. The presence of other defects such as micro-cracks, pores, and lack of fusion within the volume enhance the propagation rate. Additionally, some authors reported that inter-layer bonding is weak in SLM materials [45,46]. Therefore, SLM fabricated 304 L SS demonstrate lower fatigue strength than bulk material.

Further analysis was performed to evaluate the CSM method in identifying nucleation and propagation phase. Rough finished and notched rough finished wrought specimens were prepared to conduct fatigue test. Rough finished specimens were chosen because these are easy to prepare. Figure 15 shows the notch location in the specimen. Figure 16 illustrates the nucleation and propagation phase identified using the CSM method for the specimens. As we know, notched specimens have higher stress concentration, hence, they fail earlier. The number of cycles decreases more in nucleation phase due to the notch while the influence of notch on the propagation is minimal. This attributes to the proper implementation of the CSM method in identifying the end of nucleation phase. In future, the method will be validated determining the crack length at the end of nucleation phase.

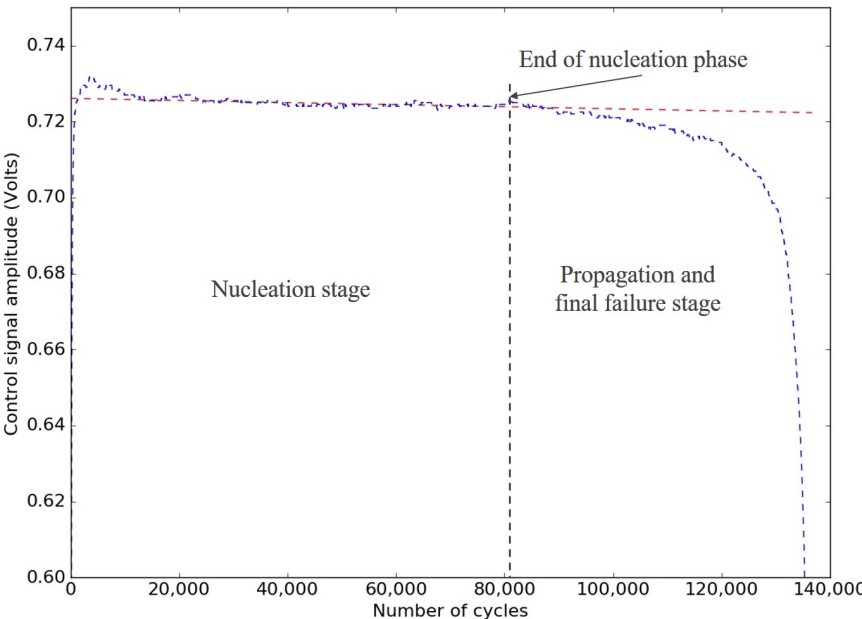

**Figure 10.** Nucleation and propagation stage of a fine finished wrought specimen displaced at 0.200 mm amplitude.

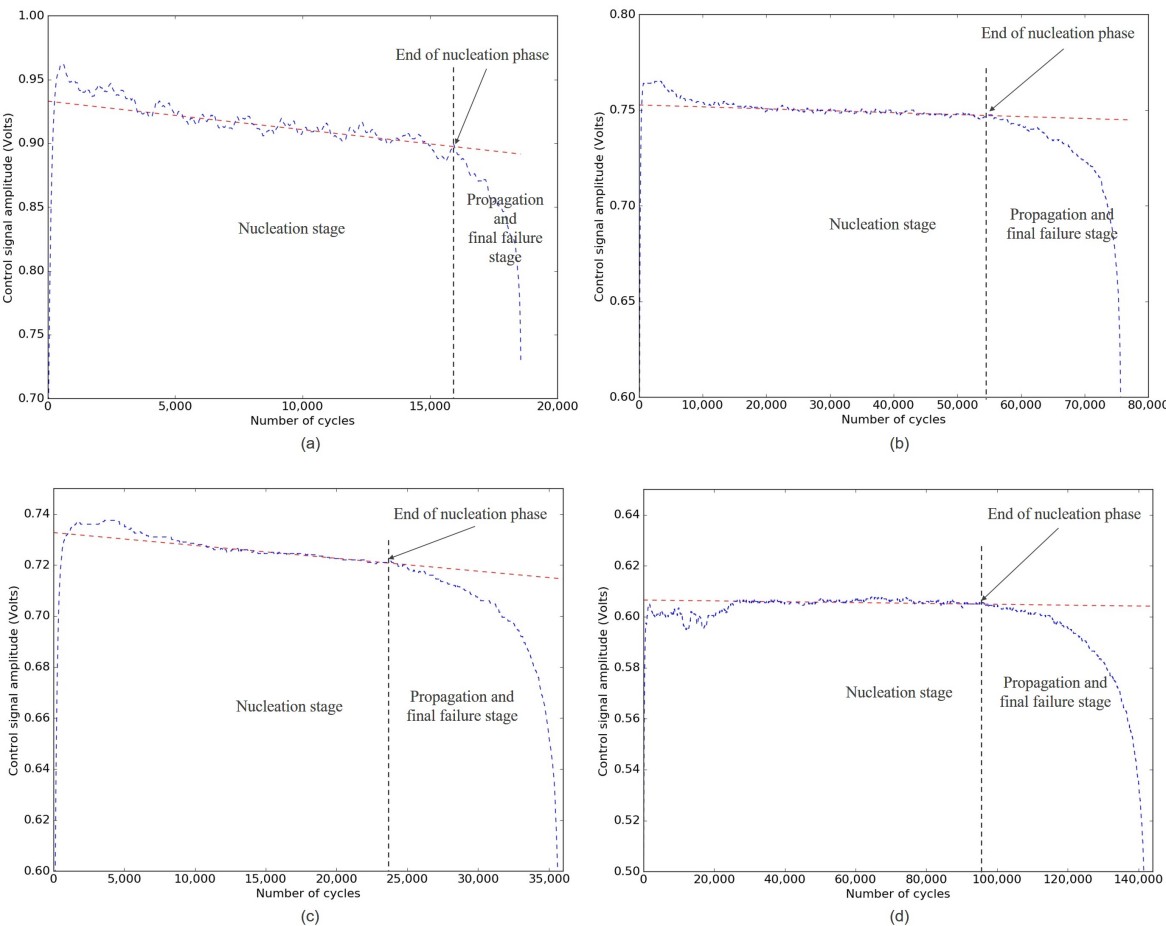

**Figure 11.** Nucleation and propagation stage of fine finished specimens, (**a**) wrought specimen displaced at 0.300 mm amplitude; (**b**) wrought specimen displaced at 0.225 mm amplitude; (**c**) SLM specimen displaced at 0.200 mm; (**d**) SLM specimen displaced at 0.175 mm amplitude.

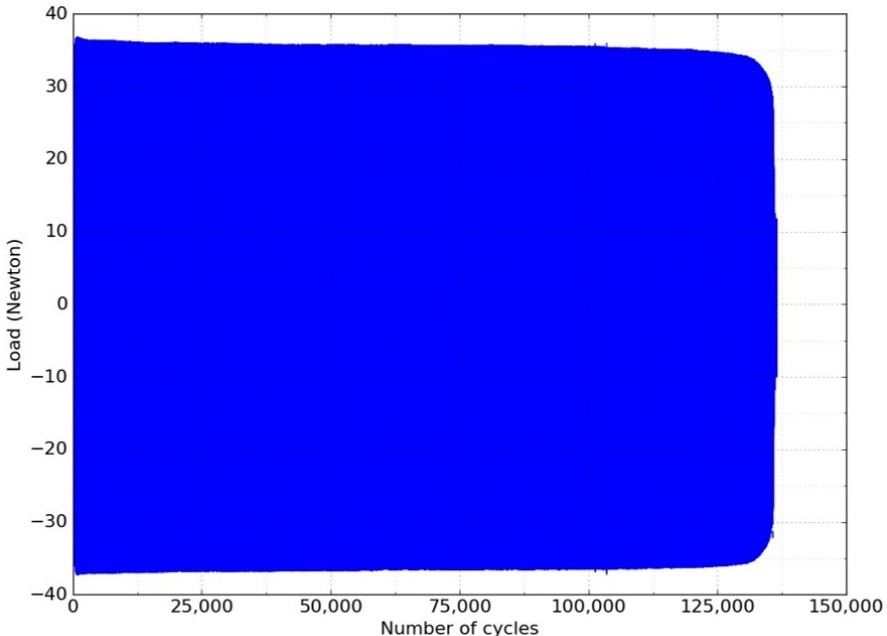

**Figure 12.** Load values for the specimen displaced at 0.200 mm amplitude.

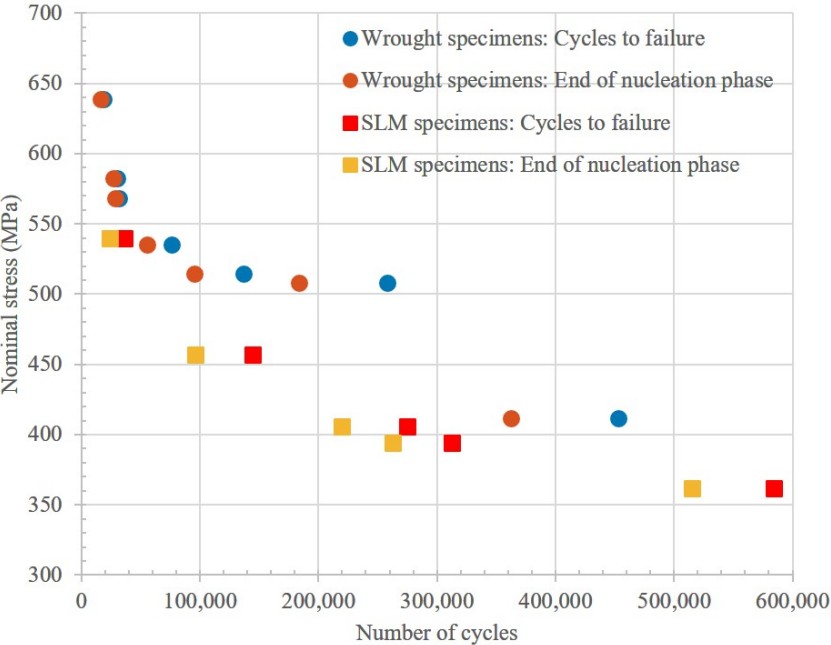

**Figure 13.** End of nucleation phase and cycles to failure for fine finished wrought and SLM fabricated specimens.

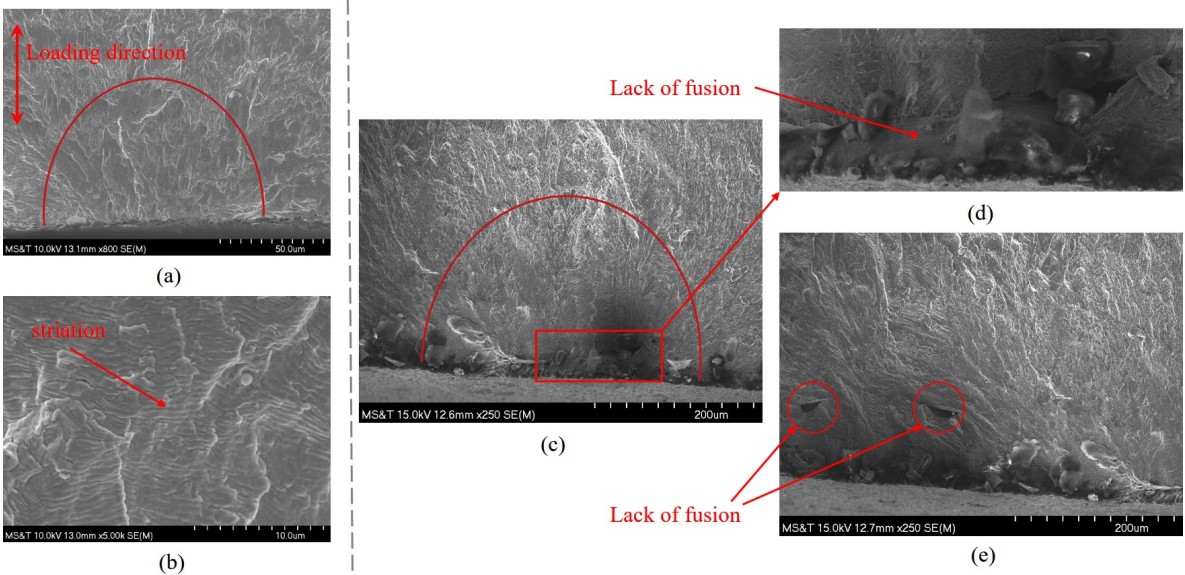

**Figure 14.** Fracture surface analysis of fine finished wrought and SLM fabricated materials, (**a**) crack nucleation site; and (**b**) crack propagation site of a wrought specimen, the nominal stress is 411.67 MPa; (**c**) crack nucleation site of SLM fabricated specimen, the nominal stress is 394.76 MPa; (**d**) presence of lack of fusion at the surface initiating the crack; (**e**) presence of lack of fusion within the volume enhancing crack propagation rate.

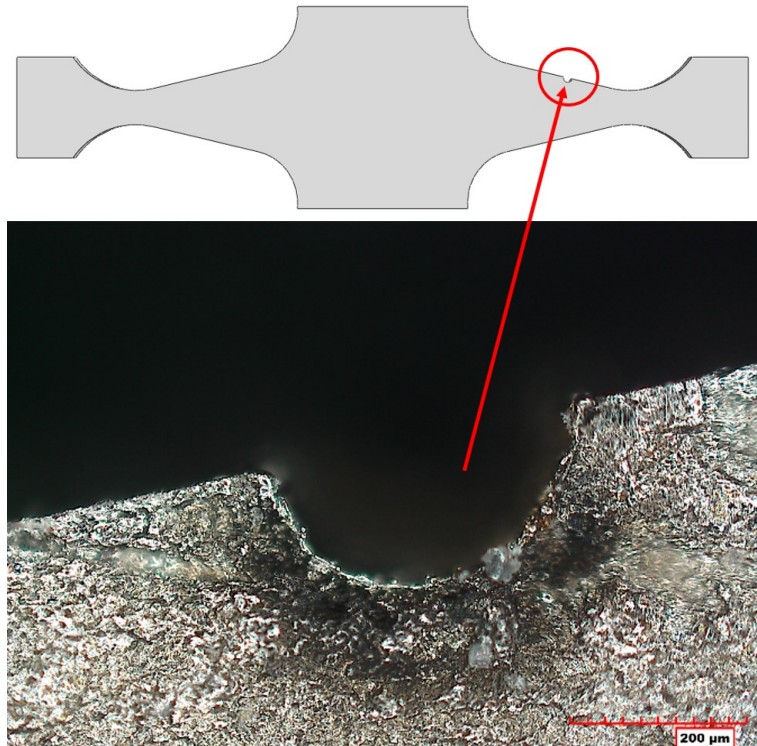

**Figure 15.** Notched rough finished wrought specimens prepared with a W-EDM wire radius of 0.125 mm. The average radius of the notch measured was 0.180 mm.

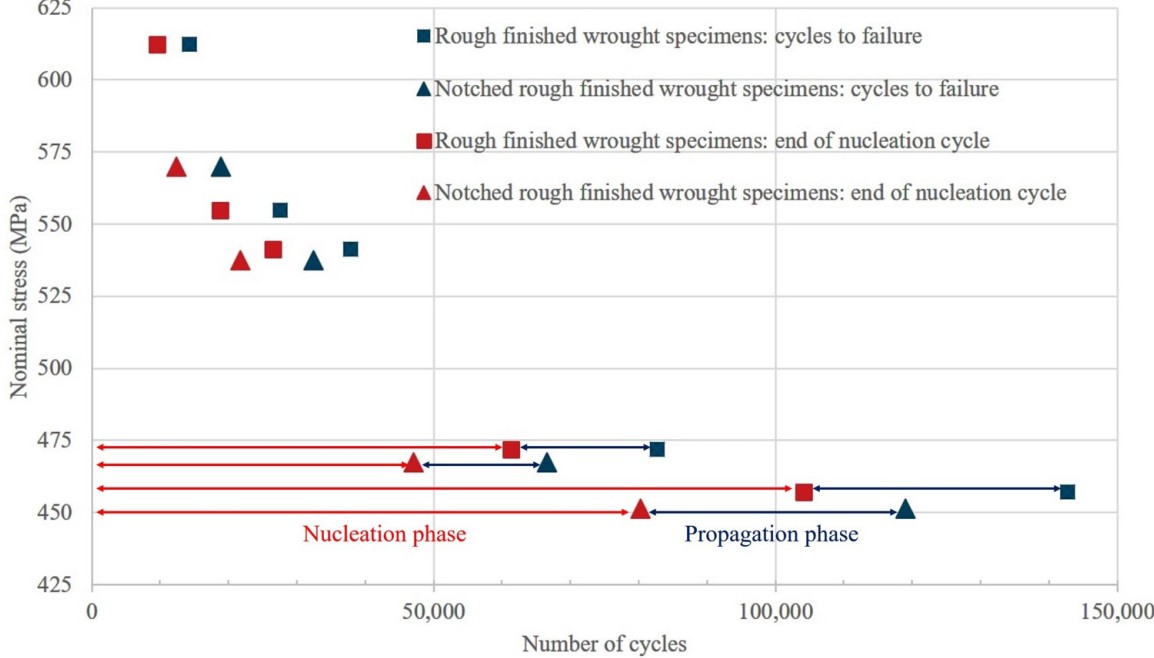

**Figure 16.** Nucleation and propagation phase of rough finished wrought and notched rough finished wrought specimens.

The Wohler curve for fine finished and rough finished wrought specimens, and fine finished SLM specimens was plotted as shown in Figure 17. Both fine finished SLM and rough finished wrought materials exhibit low fatigue strength compared to the fine finished wrought material. The endurance limit ($10^7$) of the fine finished, rough finished wrought specimen and fine finished SLM specimen reported here are 404.26, 344.62 and 336.52 MPa respectively. The yield tensile strength (YTS) and

ultimate tensile strength (UTS) for the wrought material used in this study are 582.5 and 780.9 respectively while for the SLM material with the same process parameter used, they are 368.4 and 536.7 MPa respectively [47]. These results are in good agreement with the general relationship between fatigue limit and ultimate tensile strength [43]. The endurance limit is also comparable with the results reported by other authors [38,48].

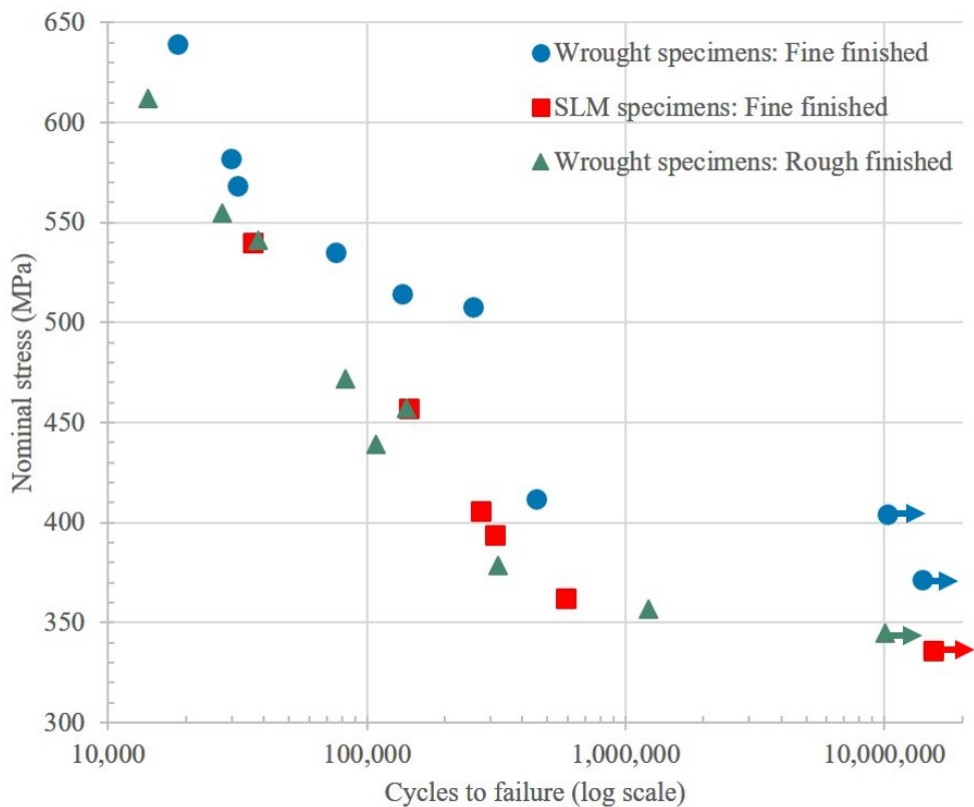

**Figure 17.** Wohler curve plot of wrought and SLM fabricated specimens.

## 6. Conclusions

In this study, a dual gauge section Krouse type mini specimen was designed to achieve a constant stress distribution with increased volume to conduct fatigue test. The test was performed with a simply supported loading mechanism on wrought and additively manufactured materials using a unique adaptive displacement controlled mini fatigue test set up. A new diagnosis method named control signal monitoring (CSM) was employed to identify the nucleation and propagation stages. The test results and analyses illustrate that SLM fabricated 304L stainless steel demonstrate lower fatigue strength in terms of both the nucleation and propagation cycles compared to bulk wrought material. The test method developed here can be applied in the extensive study on other additively manufactured materials in the future.

**Author Contributions:** The contribution of each author is summarized as conceptualization, M.M.P. and C.C.; methodology, M.M.P. and S.K.; software, M.M.P.; data curation, M.M.P.; validation M.M.P.; formal analysis, M.M.P. and Y.C.; investigation, M.M.P.; writing—original draft, M.M.P.; writing—review and editing, M.M.P., Y.C., C.C., J.W.N., F.L.; visualization, M.M.P.; supervision, J.W.N. and F.L.; project administration, F.L.; funding acquisition, F.L.

**Funding:** This research was supported by National Science Foundation Grant CMMI-1625736. Part of the work was also funded by the Department of Energy's Kansas City National Security Campus which is operated and managed by Honeywell Federal Manufacturing Technologies, LLC under contract number DE-NA0002839.

**Conflicts of Interest:** The authors declare no conflict of interest.

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
