# Peer review of "A Displacement Controlled Fatigue Test Method for Additively Manufactured Materials"

_applsci, doi:10.3390/app9163226_

Round 1
Reviewer 1 Report
Please see the attached letter.

Author Response
As authors, we would first like to express our appreciation to the reviewer to add his/her valuable comments in reviewing the manuscript. Our reply for each comment is presented below.

Reviewer 2 Report
see attached manuscript-in yellow - not clear or needs rewriting.
you may include this paper in your into
Weibull statistical analysis of Krouse type bending fatigue of nuclear materials
Ahmed S.Haidyrahc Joseph W.Newkirk Carlos H.Castañoa

Author Response
As authors, we would first like to express our appreciation to the reviewer to add his/her valuable comments in reviewing the manuscript. Our reply to each comment is presented below.
1. See attached manuscript - in yellow-not clear or needs rewriting
Line 71
Line 88
Line 235
2. You may include this paper in your into
Weibull statistical analysis of Krouse type bending fatigue of nuclear materials
Ahmed S. Haidyrah, Joseph W. Newkirk, Carlos H. Castano
Line 357
Reviewer 3 Report
The novelty seems not significant or should be expressed more explicitly. The theoretical part seems incorrect. In the methodology section, the beam model and equations are used. The authors should be noted that the beam model is based on the assumption that the size of its cross-section is much much smaller than its length. But it is obvious that the specimen in figure 1 cannot be modelled as a Bernoulli-Euler beam, therefore, equations (1) (2) (3) (4) etc. are not proper for this specimen. This specimen should be modelled based on the elasticity theory. How do you design the specimen to keep the constant stress distribution? The FEM analysis in section 3.4 and 3.5 should give the convergence analysis. In section 5 "Results and discussion", all results are from the experiment? The experimental results should be compared with the theoretical or simulation results. The errors between them should be analyzed in detail. Based on these comments, this reviewer does not support to publish this manuscript in present form.Author Response
Please see the attachment

Round 2
Reviewer 3 Report
The authors have basically revised the paper according to the reviewers' comments. The only question remains as below:
the authors claimed that previous studies have shown that simple beam equation is applicable to calculate the stress in miniature wedge-shaped specimen [34–39]. However, the reviewer doesn't find any error analysis for using the simple rectangle beam equations for this kind of specimen in these studies, although the same equations were used in [37]. The reviewer suggests the authors clarify this question. Besides, more attention should be paid to the formatting of this paper, for instance, there should be no indent in front of "where" after formula. The article can be accepted after minor revision.
